# COSFORMER : RETHINKING SOFTMAX IN ATTENTION

[1]**Zhen Qin**[†]   [1,3]**Weixuan Sun**[†]   [1,4]**Hui Deng**[†]   [3]**Dongxu Li**   [1]**Yunshen Wei**   [1]**Baohong Lv**
[1]**Junjie Yan**   [2,5]**Lingpeng Kong**   [1,2]**Yiran Zhong**[*]
[1]SenseTime Research      [2]Shanghai AI Laboratory      [3]Australian National University
[4]Northwestern Polytechnical University      [5]The University of Hong Kong
{lastnamefirstname}@sensetime.com,lpk@cs.hku.hk

## ABSTRACT

Transformer has shown great successes in natural language processing, computer vision, and audio processing. As one of its core components, the softmax attention helps to capture long-range dependencies yet prohibits its scale-up due to the quadratic space and time complexity to the sequence length. Kernel methods are often adopted to reduce the complexity by approximating the softmax operator. Nevertheless, due to the approximation errors, their performances vary in different tasks/corpus and suffer crucial performance drops when compared with the vanilla softmax attention. In this paper, we propose a linear transformer called COSFORMER that can achieve comparable or better accuracy to the vanilla transformer in both casual and cross attentions. COSFORMER is based on two key properties of softmax attention: i). non-negativeness of the attention matrix; ii). a non-linear re-weighting scheme that can concentrate the distribution of the attention matrix. As its linear substitute, COSFORMER fulfills these properties with a linear operator and a *cosine*-based distance re-weighting mechanism. Extensive experiments on language modeling and text understanding tasks demonstrate the effectiveness of our method. We further examine our method on long sequences and achieve state-of-the-art performance on the *Long-Range Arena* benchmark. The source code is available at COSFORMER .

## 1   INTRODUCTION

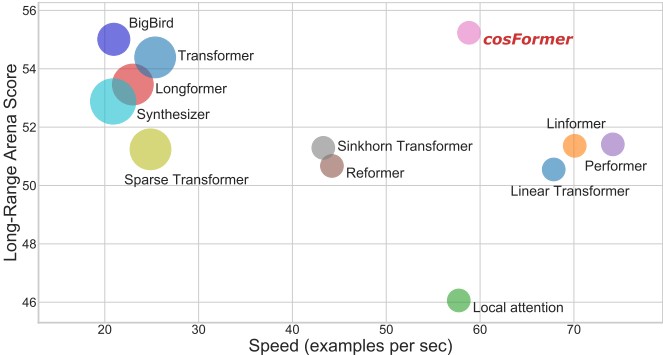

Figure 1: Performance ($y$ axis), speed ($x$ axis), and memory footprint (circle sizes) of efficient transformers on the Long-Range Arena benchmark. The proposed COSFORMER achieves an all-around supremacy over competing methods in the top left quadrant.

With years of development, the transformer model (Vaswani et al., 2017) and its variants (Zaheer et al., 2020; Wang et al., 2020; Tay et al., 2020a) have been successfully adapted to three most popular artificial intelligence (AI) fields: *i.e.,* natural language processing (Devlin et al., 2019; Liu et al., 2019), computer vision (Dosovitskiy et al., 2020; Carion et al., 2020; Liu et al., 2021) and audio processing (Schneider et al., 2019; Baevski et al., 2020). Compared with conventional

---

[*]Indicates the corresponding author. † Indicates equal contribution.

recurrent (Hochreiter & Schmidhuber, 1997) and convolutional architectures (He et al., 2016), transformer-based architectures are generally more scalable to data volumes (Brown et al., 2020) and stronger in capturing global information with less inductive bias, thus excelling on many tasks.

Dot-product attention with softmax normalization is the cornerstone of the transformer to capture long-range dependencies. However, its quadratic space and time complexity with regard to the length of the sequence make its computational overhead prohibitive, especially for long inputs. To address this issue, numerous methods are proposed recently, such as the sparse attention matrix (Zaheer et al., 2020; Beltagy et al., 2020; Tay et al., 2020a; Kitaev et al., 2019; Child et al., 2019),low-rank representations (Wang et al., 2020) or kernel-based methods (Peng et al., 2020; Choromanski et al., 2020; Katharopoulos et al., 2020), among many others. These methods achieve reduced computational complexity with comparable performances when compared with the vanilla attention architecture on several selected tasks or corpus.

However, the improved efficiency is usually achieved via introducing additional yet often impractical assumptions on the attention matrix (Wang et al., 2020) or with valid approximation of softmax operation only within constrained theoretical bounds (Choromanski et al., 2020; Peng et al., 2020) Therefore, when their assumptions are unsatisfied or when approximation errors get accumulated, these methods may not always be advantageous over the vanilla architecture (Narang et al., 2021). Consequently, performance deficiencies in a broad application spectrum are often observed in these transformer variants, especially those with linear complexity. For example, the Performer (Choromanski et al., 2020), RFA (Peng et al., 2020) and Reformer (Kitaev et al., 2019) show less satisfactory performance on the GLUE benchmark (Wang et al., 2018) when compared with the vanilla architecture as suggested in our preliminary experiments (Tab. 2). Furthermore, many of these aforementioned methods are not applicable to casual attentions, which are critical for auto-regressive training. For example, techniques proposed in Linformer (Wang et al., 2020) and BigBird (Zaheer et al., 2020) are specific to cross attentions.

Since the softmax operator appears to be the main hurdle while efficient yet accurate approximation to softmax is difficult to achieve, one question naturally arises: "*Can we replace the softmax operator with a linear function instead, while maintaining its key properties?*". By digging into the softmax attention, we find two key properties that affect its empirical performance: (i) elements in the attention matrix are non-negative (Tsai et al., 2019; Katharopoulos et al., 2020); (ii) the non-linear re-weighting scheme acts as a stabilizer for the attention weights (Titsias, 2016; Gao & Pavel, 2017; Jang et al., 2016). These findings reveal some new insights of the current approaches. For example, the linear transformer (Katharopoulos et al., 2020) achieves property (i) using an exponential linear unit (Clevert et al., 2016) activation function. However, due to lack of the re-weighting scheme, it underperforms other efficient transformer variants on the Long-Range Arena benchmark as shown in Figure 1 as well as the language modeling task (Table 2) based on our controlled experiments.

In this paper, we propose a new variant of linear transformer called COSFORMER that satisfies both of the above properties. Specifically, we enforce the non-negative property by passing the features to a ReLU (Agarap, 2018) activation function before computing the similarity scores. In this way, we encourage the model to avoid aggregating negatively-correlated contextual information. Further, we adopt a *cos* re-weighting scheme to stabilize the attention weights. This helps the model to amplify local correlations, which usually contain more relevant information for natural language tasks. Thanks to the Ptolemy's theorem, our attention can be *exactly* decomposed into a linear form. We perform extensive experiments on both autoregressive language models and bidirectional models on five public benchmarks, including WikiText-103 (Merity et al., 2017), GLUE (Wang et al., 2018), IMDB (Maas et al., 2011), AMAZON (Ni et al., 2019) and Long-Range Arena benchmark (Tay et al., 2020b). Our model shows much better inference speed and smaller memory footprint, while achieving on par performance with the vanilla transformer. It is noteworthy that our method ranks 1[st] on the Long-Range Arena benchmark, showing favorable performance than other competitors, which well demonstrates its strong capacity in modeling long sequence inputs.

## 2 OUR METHOD

In this section, we provide technique details of our linear transformer called COSFORMER . The key insight of the COSFORMER is to replace the non-decomposable non-linear softmax operation by a linear operation with decomposable non-linear re-weighting mechanism. Our model is applicable

to both casual and cross attentions with a linear time and space complexity with regard to the input sequence length, thus exhibiting strong capacity in modeling long-range dependencies.

## 2.1 THE GENERAL FORM OF TRANSFORMER

Given an input sequence $x$ with length of $N$, we first represent it in the embedding space $x \in \mathbb{R}^{N \times d}$ with feature dimension of $d$. A transformer block $\mathcal{T} : \mathbb{R}^{N \times d} \to \mathbb{R}^{N \times d}$ with input $x$ is defined as:

$$\mathcal{T}(x) = \mathcal{F}(\mathcal{A}(x) + x), \tag{1}$$

where $\mathcal{F}$ is a feedforward network that contains a residual connection; $\mathcal{A}$ is the self-attention function that computes the attention matrix $A$, which has quadratic space and time complexity with respect to $N$, thus becoming the computation bottleneck of $\mathcal{T}$ on long inputs.

There are three key components in $\mathcal{A}$, namely, *query* $(Q)$, *key* $(K)$, *value* $(V)$ computed through three learnable linear matrices $W_Q, W_K, W_V$: $Q = xW_Q, K = xW_K, V = xW_V$. We use $M_i$ to represent the i-th row of a matrix $M$, then the output $\mathcal{O} \in \mathbb{R}^{N \times d}$ of $\mathcal{A}(x)$ can be computed as:

$$\mathcal{O} = \mathcal{A}(x) = [\mathcal{O}_1, \dots, \mathcal{O}_N]^T, \quad \mathcal{O}_i = \sum_j \frac{\mathcal{S}(Q_i, K_j)}{\sum_j \mathcal{S}(Q_i, K_j)} V_j, \tag{2}$$

where $\mathcal{S}(\cdot)$ measures the similarity between queries. If $\mathcal{S}(Q, K) = \exp(QK^T)$, the Eq. 2 becomes the dot-product attention with softmax normalization. In this case, the space and time complexity to compute one row of the output $\mathcal{O}_i$ is $O(N)$. Therefore, the total space and time complexity for computing $\mathcal{O}$ grows quadratically with respect to the input length.

## 2.2 LINEARIZATION OF SELF-ATTENTION

According to Eq. 2, we can select any similarity functions to compute the attention matrix. In order to maintain a linear computation budget, one solution is to adopt a decomposable similarity function such that:

$$\mathcal{S}(Q_i, K_j) = \phi(Q_i)\phi(K_j)^T, \tag{3}$$

where $\phi$ is a kernel function that maps the queries and keys to their hidden representations. Then one can rewrite Eq. 2 in the form of kernel functions as:

$$O_i = \frac{\sum_{j=1}^{N}(\phi(Q_i)\phi(K_j)^T)V_j}{\sum_{j=1}^{N}(\phi(Q_i)\phi(K_j)^T)}, \tag{4}$$

After that, attention operation in linear complexity is achieved via the matrix product property:

$$(\phi(Q)\phi(K)^T)V = \phi(Q)(\phi(K)^T V). \tag{5}$$

In this form (Eq. 5), instead of explicitly computing the attention matrix $A = QK^T \in \mathbb{R}^{N \times N}$, we calculate the $\phi(K)^T V \in \mathbb{R}^{d \times d}$ first, and then multiplying $\phi(Q) \in \mathbb{R}^{N \times d}$. By using this trick, we only incurs a computation complexity of $O(Nd^2)$. Note that in typical natural language tasks, the feature dimension of one head $d$ is always much smaller than the input sequence length $N$ ($d \ll N$), so we can safely omit $d$ and achieve computation complexity of $O(N)$, as illustrated in Figure 2.

**Previous Solutions** As aforementioned, the key to the linear attentions is to find a decomposable similarity function $\mathcal{S}(\cdot)$ that generalizes well to different tasks. Most existing linear transformers are trying to find an unbiased estimation of the softmax attention. For example, RFA (Peng et al., 2020) approximates the softmax operation with random feature maps using theorem of random fourier features (Rahimi & Recht, 2008) and the Performer (Choromanski et al., 2020) utilizes positive random features to approximate it. However, we empirically find that these methods are sensitive to the selection of sampling rate and becomes unstable if the sampling rate gets too high. Also, to accommodate recency bias, gating mechanisms are employed to better exploit more recent context.

Another group of works attempt to directly replace the softmax with a linear operation. For example, the linear transformer (Katharopoulos et al., 2020) model replaces the softmax similarity function with a pure dot product $\mathcal{S} = QK^T$, and use a non-linear activation function $\phi(\cdot) = \mathrm{elu}(\cdot) + 1$ to model the pairwise relation between features. However, our controlled experiments show that their solution does not necessarily generalize well on many downstream tasks (Tab. 2) or the Long-Range Arena benchmark (Tab. 4). In this paper, we propose a new replacement of softmax that not only achieves comparable or better performance than the softmax attention in a wide range of tasks, but also enjoys linear space and time complexity.

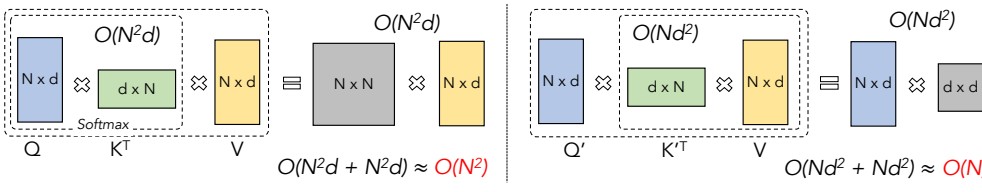

Figure 2: Illustration of the computations for vanilla self attention (left) and linearized attention (right). The input length is $N$ and feature dimension is $d$, with $d \ll N$. Tensors in the same box are associated for computation. The linearized formulation allows $O(N)$ time and space complexity.

## 2.3 ANALYSIS OF SOFTMAX ATTENTION

In the vanilla transformer architecture, when $\mathcal{S}(Q, K) = \exp(QK^T)$, the softmax operation is applied to obtain row-wise normalization on the attention matrix $A \in \mathbb{R}^{N \times N}$ as shown in the Eq. 2. In other words, we normalize the relations of each element in the input sequence to all other elements in order to obtain a weighted aggregation of contextual information. However, apart from the good empirical performance of softmax attention, what are the crucial and necessary characteristics of it remain only loosely determined in the original transformer paper and follow-up works.

In this work, we empirically identify two key properties of the softmax operation that may play important roles for its performance: 1) it ensures all values in the attention matrix $A$ to be non-negative; 2) it provides a non-linear re-weighting mechanism to concentrates the distribution of attention connections and stabilizes the training(Titsias, 2016; Gao & Pavel, 2017; Jang et al., 2016).

To validate these assumptions, we design the following preliminary studies as shown in Table 1. First, to validate the importance of non-negativity, we compare three instantiations of

Table 1: Analysis of the softmax properties. All attention variants are implemented in the RoBERTa (Liu et al., 2019) architecture and are pre-trained on the WikiText-103 (Merity et al., 2017) dataset. The Loss represents the validation loss. We then fine-tune these variants on each downstream datasets and show the accuracy (the higher the better).

|  | Loss | QQP | SST-2 | MNLI |
|---|---|---|---|---|
| $\phi_{\mathbf{I}}$ | 2.343 | 84.23 | 76.26 | 58.27 |
| $\phi_{\mathrm{LeakyReLU}}$ | 2.246 | 84.46 | 78.21 | 74.26 |
| $\phi_{\mathrm{ReLU}}$ | 1.993 | 88.86 | 89.90 | 77.86 |
| softmax | 1.915 | 88.41 | 92.31 | 79.15 |

the function $\phi$ in equation 3: an identify mapping $\phi_{\mathbf{I}} = \mathbf{I}$ that does not preserve the non-negativity, and the other variant $\phi_{\mathrm{ReLU}(\cdot)} = \mathrm{ReLU}(\cdot)$ that retains only positive input values while replacing negative values to zeros. We also add the $\phi_{\mathrm{LeakyReLU}(\cdot)} = \mathrm{LeakyReLU}(\cdot)$ variant as it does not have the non-negativity as well but have the same non-linearly as the ReLU one. Second, to demonstrate the effect of non-linear re-weighting, we compare the models using only $\phi_{\mathrm{ReLU}(\cdot)}$ without any re-weighting and those with softmax operations. From Table 1, the superior results of $\phi_{\mathrm{ReLU}}$ over $\phi_{\mathbf{I}}$ and $\phi_{\mathrm{LeakyReLU}}$ demonstrate the benefit of retaining non-negative values. Our conjecture is that by retaining only positive values in the similarity matrices, the model ignores features with negative correlations, thus effectively avoiding aggregating irrelevant contextual information. By comparing the results of $\phi_{\mathrm{ReLU}}$ with the softmax, we observe that models with softmax re-weighting converge faster and generalize better to downstream tasks. This might be explained as softmax normalization amplifies the correlated pairs, which might be useful to identify useful patterns.

## 2.4 COSFORMER

Based on the observations above, we propose our model COSFORMER, which discards entirely the softmax normalization while still features the non-negativity and re-weighting mechanism. Our COSFORMER consists two main components: a linear projection kernel $\phi_{\mathrm{linear}}$ and a *cos*-Based Re-weighting mechanism. Below we describe details of each components:

**Linear projection kernel $\phi_{\mathrm{linear}}$** Recall the general form of the attention in Eq. 2, let us define a linear similarity as:

$$\mathcal{S}(Q, K) = \mathrm{s}(\phi_{\mathrm{linear}}(Q), \phi_{\mathrm{linear}}(K)) = \mathrm{s}(Q', K') \tag{6}$$

where $\phi_{\mathrm{linear}}$ is the transformation function that map queries $Q$ and keys $K$ to our desired representations $Q'$ and $K'$, and s is a function that can be linearly decomposed to measure the similarity between $Q'$ and $K'$. Specifically, in order to ensure a full positive attention matrix $A$ and avoid

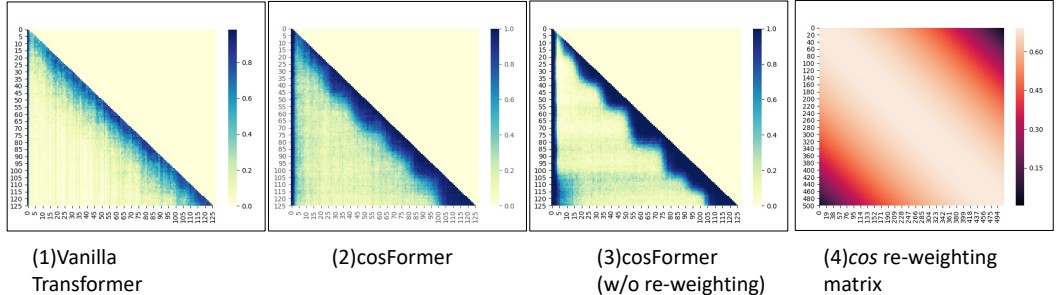

(1)Vanilla Transformer | (2)cosFormer | (3)cosFormer (w/o re-weighting) | (4)*cos* re-weighting matrix

Figure 3: (1): Attention matrix of vanilla transformer.(2):Attention matrix of COSFORMER .(3): Attention matrix of COSFORMER without re-weighting. (4): Visualization of the *cos*-based distance matrix. After re-weighting, we can see a smoother attention distribution along the diagonal region of attention matrix, exhibiting a similar pattern to the vanilla transformer, which assists to stabilize the training.

aggregating negatively-correlated information, we adopt $\text{ReLU}(\cdot)$ as the transformation functions and therefore effectively eliminate negative values:

$$\phi_{\text{linear}}(x) = \text{ReLU}(x) \tag{7}$$

As $Q'$ and $K'$ contain only non-negative values, we directly take their dot-product $s(x,y) = xy^T, x, y \in \mathbb{R}^{1\times d}$ followed by a row-wise normalization to compute attention matrices:

$$\mathcal{O}_i = \frac{\sum_{j=1}^N f(\phi_{\text{linear}}(Q_i), \phi_{\text{linear}}(K_j))V_j}{\sum_{j=1}^N f(\phi_{\text{linear}}(Q_i), \phi_{\text{linear}}(K_j))} = \frac{\sum_{j=1}^N (\text{ReLU}(Q_i)\text{ReLU}(K_j)^T)V_j}{\sum_{j=1}^N (\text{ReLU}(Q_i)\text{ReLU}(K_j)^T)} \tag{8}$$

Based on Eq. 4, we rearrange the order of dot-product and obtain the formulation of the proposed attention in linear complexity as:

$$\mathcal{O}_i = \frac{\text{ReLU}(Q_i)\sum_{j=1}^N \text{ReLU}(K_j)^T V_j}{\text{ReLU}(Q_i)\sum_{j=1}^N \text{ReLU}(K_j)^T} \tag{9}$$

***cos*-Based Re-weighting Mechanism**   The non-linear re-weighting mechanism introduced by the softmax attention can concentrate the distribution of the attention weights and therefore stabilize the training process (Titsias, 2016; Gao & Pavel, 2017; Jang et al., 2016). We also empirically find that it can punish far-away connections and enforce locality in some cases. In fact, such locality bias, *i.e.,* a large portion of contextual dependencies are from neighboring tokens, is commonly observed on downstream NLP tasks (Clark et al., 2019; Kovaleva et al., 2019), as shown in Figure 3 (1).

Based on the assumption above, what we need to fulfill the second property of softmax may be a decomposable re-weighting mechanism that can introduce recency bias to the attention matrix. Here, we propose a *cos*-based re-weighting mechanism as it perfectly fit our purpose: 1). the Ptolemy's theorem ensures the cos weights can be decomposed into two summations; 2). as shown in Figure 3 (4), the cos will put more weights on the neighbouring tokens and therefore enforces locality. Also, by comparing the attention matrices in Figure 3 (2) and (3), the COSFORMER enforces more locality than the one without the re-weighting mechanism.

Specifically, by combining with Eq 6, the model with cosine re-weighting is defined as:

$$s(Q_i', K_j') = Q_i' K_j'^T \cos\left(\frac{\pi}{2} \times \frac{i-j}{M}\right) \tag{10}$$

By leveraging the Ptolemy's theorem, we decompose this formulation as:

$$Q_i' K_j'^T \cos\left(\frac{\pi}{2} \times \frac{i-j}{M}\right) = Q_i' K_j'^T \left(\cos\left(\frac{\pi i}{2M}\right)\cos\left(\frac{\pi j}{2M}\right) + \sin\left(\frac{\pi i}{2M}\right)\sin\left(\frac{\pi j}{2M}\right)\right)$$

$$= \left(Q_i'\cos\left(\frac{\pi i}{2M}\right)\right)\left(K_j'\cos\left(\frac{\pi j}{2M}\right)\right)^T + \left(Q_i'\sin\left(\frac{\pi i}{2M}\right)\right)\left(K_j'\sin\left(\frac{\pi j}{2M}\right)\right)^T$$

where $i, j = 1, ..., N, M \geq N$, and $Q' = \text{ReLU}(Q), K' = \text{ReLU}(K)$. Let $Q_i^{\cos} = Q_i'\cos\left(\frac{\pi i}{2M}\right)$, $Q_i^{\sin} = Q_i'\sin\left(\frac{\pi i}{2M}\right)$, $K_j^{\cos} = K_j'\cos\left(\frac{\pi j}{2M}\right)$, $K_j^{\sin} = K_j'\sin\left(\frac{\pi j}{2M}\right)$, the output of the proposed attention module can be expressed as:

$$O_i = \frac{\sum_{j=1}^N f(Q_i', K_j')V_j}{\sum_{j=1}^N f(Q_i', K_j')} = \frac{\sum_{j=1}^N Q_i^{\cos}\left((K_j^{\cos})^T V_j\right) + \sum_{j=1}^N Q_i^{\sin}\left((K_j^{\sin})^T V_j\right)}{\sum_{j=1}^N Q_i^{\cos}(K_j^{\cos})^T + \sum_{j=1}^N Q_i^{\sin}(K_j^{\sin})^T}, \tag{11}$$

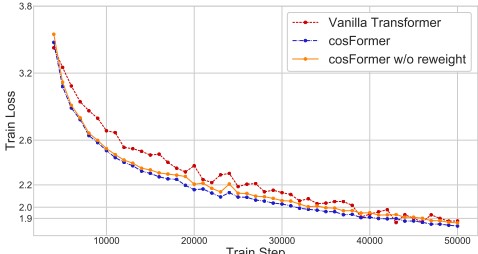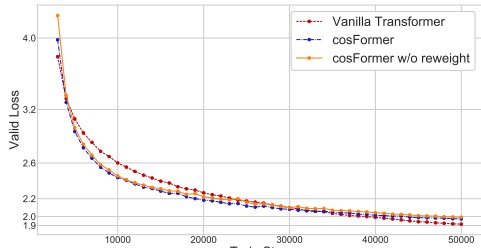

Figure 4: Training loss (left) and validation loss (right) of the bidirectional language modeling pre-train. In both training and validation, the proposed COSFORMER has a faster converge speed than vanilla transformer.

where $O_i$ is the output at the $i^{th}$ position of the sequence from the attention module. Detailed derivation are included in the Appendix. Without losing the generality, our method achieves a linear complexity as:

$$\mathcal{O} = \mathcal{S}(Q, K)V = (Q^{\cos}K^{\cos} + Q^{\sin}K^{\sin})V = Q^{\cos}(K^{\cos}V) + Q^{\sin}(K^{\sin}V) \quad (12)$$

**Relation to positional encoding.** COSFORMER can be seen as a new way of introducing the relative positional bias to the efficient transformer. Compared with the Rotary Position Embedding (Su et al., 2021), they use a more complex position embedding strategy and did not enforce the non-negativity to the similarity scores as ours. Also, since they only change the position embedding on the numerator while keeping the denominator unchanged, the summation of their attention scores is not equal to 1. For Stochastic Positional Encoding (Liutkus et al., 2021), they use a sampling strategy to approximate the softmax, and introduce relative positional encoding to linear transformers.

## 3 EXPERIMENTS

In this section, we experimentally validate the effectiveness of the proposed method in multiple settings. The purposes of the experiments are three-fold. First, we validate the capacity of COSFORMER in language modeling through autoregressive (Sec. 3.1) and bidirectional (Sec. 3.2) setups using WikiText-103 (Merity et al., 2017). In this way, we validate the effectiveness of the proposed linear attention module in both causal and non-causal cases. Second, we investigate the generalization ability of COSFORMER on downstream tasks by comparisons with other existing transformer variants. This is achieved by performing comparative finetuning experiments on five datasets, including GLUE (QQP, SST-2, MNLI) (Wang et al., 2018), IMDB (Maas et al., 2011) and AMAZON (Ni et al., 2019) (Sec. 3.3). We further compare COSFORMER with other transformer variants on the long-range-arena benchmark (Tay et al., 2020b) to understand its ability in modeling long-range dependencies (Sec. 3.4) and show comparative analysis into model efficiency (Sec. 3.5). Third, we conduct ablation studies to understand each component in COSFORMER (Sec. 3.6).

### 3.1 AUTOREGRESSIVE LANGUAGE MODELING

In autoregressive or left-to-right language modeling, we estimate the probability distribution of a token given its previous tokens. We use (Baevski & Auli, 2018) as our baseline model. Specifically, we adopt their large model which has 16 cascaded layers with a projected dimensions of 1024, and replace the self-attention module with our proposed linear attention module. We train our model on 8 Nvidia Tesla A100 GPUs with a sequence length of 512 for 150K updates on the WikiText-103 (Merity et al., 2017) and report perplexity on the validation and test splits in Table 2.

We observe that although the baseline model is a powerful standard transformer which requires quadratic computation complexity, COS-FORMER outperforms it with a clear margin in linear computation complexity. Besides, we achieve comparable perplexity to other methods on the validation set, and significantly outperform all competing methods on the test set by a clear gap, which further demonstrates the effectiveness of COSFORMER .

Table 2: Perplexity (lower is better) results of language modeling pre-training task on validation set and test set of the WikiText-103 (Merity et al., 2017) dataset.

|  | ppl(val) $\downarrow$ | ppl(test) $\downarrow$ |
| --- | --- | --- |
| Vanilla Transformer | 24.5 | 26.2 |
| Linear Transformer | 28.7 | 30.2 |
| RFA-Gaussian | 25.8 | 27.5 |
| RFA-across | 26.4 | 28.1 |
| RFA-Gate-across | 24.8 | 26.3 |
| RFA-Gate-Gaussian | **23.2** | 25.0 |
| **COSFORMER** | 23.5 | **23.1** |

Table 3: Results on fine-tuned downstream tasks based on pre-trained bidirectional model. Best result is in boldface and second best is underlined. The proposed COSFORMER achieves superb performances over competing efficient transformers and is approaching vanilla transformer.

| | QQP ↑ | SST-2 ↑ | MNLI ↑ | IMDB ↑ | AMAZON ↑ | Avg ↑ |
|---|---|---|---|---|---|---|
| Vanilla Transformer (Liu et al., 2019) | 88.41 | 92.31 | 79.15 | 92.86 | 75.79 | 85.70 |
| Performer (Choromanski et al., 2020) | 69.92 | 50.91 | 35.37 | 60.36 | 64.84 | 56.28 |
| Reformer (Kitaev et al., 2019) | 63.18 | 50.92 | 35.47 | 50.01 | 64.28 | 52.77 |
| Linear Trans. (Katharopoulos et al., 2020) | 74.85 | 84.63 | 66.56 | 91.48 | 72.50 | 78.00 |
| Longformer (Beltagy et al., 2020) | 85.51 | 88.65 | **77.22** | 91.14 | 73.34 | 83.17 |
| RFA (Peng et al., 2020) | 75.28 | 76.49 | 57.6 | 78.98 | 68.15 | 71.30 |
| **COSFORMER** | **89.26** | **91.05** | 76.70 | **92.95** | **76.30** | **85.25** |

## 3.2 BIDIRECTIONAL LANGUAGE MODEL

For bidirectional language modeling, we adopt RoBERTa (Liu et al., 2019) as the baseline model. Similarly, we replace the self-attention module in the RoBERTa by the proposed linear attention module, and keep other structures unchanged. We train this bidirectional task on 2 Nvidia Tesla A100 GPUs for 50K iterations with a input sequence length 512. As shown in Figure 4, COS-FORMER converges faster than vanilla transformer on both training and validation sets with a comparable or smaller loss values, despite it only consumes linear space and time computation complexity. In addition, the COSFORMER variant with re-weighting mechanism has both notably better converge speed and final results over the counterpart without re-weighting, which further validates the effectiveness of our *cos*-based distance matrix and also demonstrates the effectiveness of recency bias on natural language data.

## 3.3 DOWNSTREAM FINE-TUNING TASKS

In this section, we fine-tune the pre-trained model on downstream tasks to demonstrate the generalization ability of COSFORMER on downstream tasks. We use the pre-trained bidirectional model and fine-tune it on three downstream text classification tasks: GLUE (QQP, SST-2, MNLi) (Wang et al., 2018), IMDB (Maas et al., 2011) and AMAZON (Ni et al., 2019). For fair comparison, we first pre-train all the competing efficient transformer variants for the same 50K iterations on WikiText-103 (Merity et al., 2017) under the same setting, then we follow the same fine-tuning protocol as RoBERTa (Liu et al., 2019) to fine-tune these methods on the downstream tasks. From Table 3, we can see that COSFORMER outperforms baseline (Liu et al., 2019) on three out of five datasets, and achieves either best or secondary place on all five downstream datasets compared to competing efficient transformers. It is worth noting that despite Longformer (Beltagy et al., 2020) achieves better results on MNLI than COSFORMER, it requires a computation complexity of $O(Nw)$, where $w$ is window size. As shown in Figure 1, Longformer is slower and requires more memory overhead than COSFORMER. Other competing methods(Peng et al., 2020; Choromanski et al., 2020; Kitaev et al., 2019) are all based on kernel functions and have substantial performance gaps compared with our model. This validates the effectiveness of the proposed COSFORMER model compared with other efficient transformer variants.

## 3.4 RESULTS ON LONG-RANGE-ARENA BENCHMARK

To further evaluate the generalization ability of the proposed method, we train our model from scratch on Long-range-arena benchmark 2020b. Long-range-arena (Tay et al., 2020b) is a benchmark specifically designed for efficient transformers with long input sequences, thus serving as a suitable testbed to assess the quality of efficient transformer variants comparatively. To ensure fair comparison, we first implement our method on Jax (Bradbury et al., 2018), then carefully follow their preprocessing, data split, model structure and training protocol. We evaluate our method on a variety of tasks including Long sequence ListOps (Nangia & Bowman, 2018), Byte-level text classification (Maas et al., 2011), document retrieval using the ACL Anthology Network (Radev et al., 2013), image classification on sequence of pixels on CIFAR-10 (Krizhevsky & Hinton, 2009), and Pathfinder (Linsley et al., 2018). As shown in Table 4, COSFORMER overall achieves competitive results across all the tasks while achieving best performance on ListOps and Document Retrieval. For the Pathfinder task, since the distance between the two points can be very far from each other, our introduced locality bias would have negative impact to this task and show a bit lags to other SOTA methods, despite that the performance gap between our method and the vanilla transformer is small It is worth mentioning that COSFORMER achieves the best overall scores on Long-range-arena benchmark, being one of the only two models that surpass vanilla transformer architecture.

Table 4: Results on Long-range-arena benchmark. Best result is in boldface and second best is underlined. COSFORMER achieves the best average score across 5 different tasks.

| Model | ListOps ↑ | Text↑ | Retrieval↑ | Image↑ | Pathfinder↑ | Avg ↑ |
|---|---|---|---|---|---|---|
| Local Attention (Tay et al., 2020b) | 15.82 | 52.98 | 53.39 | 41.46 | 66.63 | 46.06 |
| Linear Trans. (Katharopoulos et al., 2020) | 16.13 | **65.9** | 53.09 | 42.34 | 75.3 | 50.55 |
| Reformer (Kitaev et al., 2019) | 37.27 | 56.1 | 53.4 | 38.07 | 68.5 | 50.67 |
| Sparse Trans.(Child et al., 2019) | 17.07 | 63.58 | 59.59 | **44.24** | 71.71 | 51.24 |
| Sinkhorn Trans.(Tay et al., 2020a) | 33.67 | 61.2 | 53.83 | 41.23 | 67.45 | 51.29 |
| Linformer(Wang et al., 2020) | 35.7 | 53.94 | 52.27 | 38.56 | 76.34 | 51.36 |
| Performer(Choromanski et al., 2020) | 18.01 | 65.4 | 53.82 | 42.77 | **77.05** | 51.41 |
| Synthesizer (Tay et al., 2021) | 36.99 | 61.68 | 54.67 | 41.61 | 69.45 | 52.88 |
| Longformer(Beltagy et al., 2020) | 35.63 | 62.85 | 56.89 | 42.22 | 69.71 | 53.46 |
| Transformer (Vaswani et al., 2017) | 36.37 | 64.27 | 57.46 | 42.44 | 71.4 | 54.39 |
| BigBird (Zaheer et al., 2020) | 36.05 | 64.02 | 59.29 | 40.83 | 74.87 | 55.01 |
| **COSFORMER** | **37.9** | 63.41 | **61.36** | 43.17 | 70.33 | **55.23** |

Table 5: Speed comparison on the long-range-arena benchmark in both training and inference varying sequence lengths (1-4k). We mark it with a cross if a method runs out of memory. The higher, the better.

| model | Inference Speed(steps per second)↑ | | | | Train Speed(steps per second)↑ | | | |
|---|---|---|---|---|---|---|---|---|
| | 1K | 2K | 3K | 4k | 1K | 2K | 3K | 4K |
| Transformer(Vaswani et al., 2017) | 25.37 | 7.83 | ✗ | ✗ | 6.95 | 2.23 | ✗ | ✗ |
| Local Attention(Tay et al., 2020b) | 57.73 | 33.19 | 23.36 | 17.79 | 13.45 | 6.71 | 4.32 | 3.09 |
| Linformer(Wang et al., 2020) | 70.09 | 39.1 | 27.05 | 20.62 | 14.75 | 7.09 | 4.52 | 3.21 |
| Reformer(Kitaev et al., 2019) | 44.21 | 21.58 | 12.74 | 8.37 | 11.58 | 4.98 | 2.94 | 1.95 |
| Sinkhorn Trans. (Tay et al., 2020a) | 43.29 | 23.58 | 16.53 | 12.7 | 11.09 | 5.57 | 3.68 | 2.68 |
| Synthesizer (Tay et al., 2021) | 20.89 | 6.24 | ✗ | ✗ | 6.36 | 2.01 | ✗ | ✗ |
| BirBird (Zaheer et al., 2020) | 20.96 | 11.5 | 8.12 | 6.15 | 6.46 | 3.2 | 2.13 | 1.53 |
| Linear Trans. (Katharopoulos et al., 2020) | 67.85 | 38.24 | 26.28 | 19.98 | 11.86 | 5.54 | 3.53 | 2.56 |
| Performer (Choromanski et al., 2020) | 74.15 | 42.31 | 29.5 | 22.44 | 14 | 6.49 | 4.1 | 2.94 |
| Longformer (Beltagy et al., 2020) | 22.99 | 6.72 | ✗ | ✗ | 4.4 | 1.3 | ✗ | ✗ |
| Sparse Trans. Child et al. (2019) | 24.87 | 7.5 | ✗ | ✗ | 6.77 | 2.2 | ✗ | ✗ |
| **COSFORMER** | 58.82 | 33.45 | 22.77 | 17.42 | 12.27 | 5.72 | 3.62 | 2.64 |

## 3.5 EFFICIENCY COMPARISON

In this section, we compare the efficiency of COSFORMER with other models, with a focus on long sequences as inputs. With the proposed linear attention module, we expect that COSFORMER scales comparably with other linear variants while significantly surpassing the vanilla transformer architecture. For a fair and comprehensive comparison, we implement our method and competing methods on Jax (Bradbury et al., 2018). We use the byte-level text classification benchmark and report runtime speed during both training and inference under different sequence lengths (1k-4k). We conduct experiments on one Nvidia A6000 GPU and also report the corresponding inference-time memory foot prints as shown in Figure 1. As shown in Table 5 and Figure 1, most pattern based methods (Beltagy et al., 2020; Zaheer et al., 2020; Tay et al., 2020a; 2021) and vanilla transformer (Vaswani et al., 2017) are much slower and require greater memory than COSFORMER prevents them from extending to longer sequence. Further, the kernel based methods like (Narang et al., 2021; Choromanski et al., 2020; Tay et al., 2020a) have comparable speed and memory overheads, but their performances are less satisfactory compared to COSFORMER across above metrics. In summary, our model COSFORMER achieves overall better efficiency than other linear variants while maintain superior modeling and generalization ability.

## 3.6 ABLATION: *cos*-BASED RE-WEIGHTING

By introducing *cos*-based re-weighting, we provide a non-linear mechanism to concentrate the distribution of attention connections and stabilizes the training. In this way, we encourage the model to better take into account the locality inductive biases commonly observed on many natural language tasks. In particular, we investigate the effect of the *cos*-based re-weighting in two aspects. First, as shown in Figure 4, by adding

Table 6: Performance comparison of COSFORMER with and without *cos*-based re-weighting ($\phi_{\mathrm{ReLU}}$). We evaluate on two compositive metrics. Bidirectional finetune$_{\mathrm{avg}}$: average score across 5 datasets reported in Table 3. LRA$_{\mathrm{avg}}$: average score across 5 tasks reported in Table 4.

| Model | Bidirectional finetune$_{\mathrm{avg}}$ ↑ | LRA$_{\mathrm{avg}}$ ↑ |
|---|---|---|
| $\phi_{\mathrm{ReLU}}$ | 85.12 | 54.20 |
| COSFORMER | **85.25** | **55.23** |

*cos*-based re-weighting, we obtain both notably better converge speed and final results in autoregressive language modeling. Further, in Table 6, we present a comparison between COSFORMER models with and without re-weighting mechanism. We use two composite metrics which comprehensively include 10 different datasets from bidirectional downstream fine-tuning tasks and long-range-arena (Tay et al., 2020b). COSFORMER achieves overall better results over the counterpart without re-weighting, improving the average scores on bidirectional finetuning and long-range-arena by a clear margin. This verifies that the proposed re-weighting effectively incorporates the locality inductive biases for natural language tasks.

## 4 RELATED WORK

This section will introduce the existing works on improving the efficiency of Transformers, they can be broadly divided into two categories, *Pattern based methods* and *Kernel based methods*.

**Pattern based method**   Pattern based methods sparsify the attention matrix with handcrafted or learnable patterns. As an early approach, Lee et al. (2019) leverages the inducing points from the sparse Gaussian process to reduce the quadratic complexities of a transformer. Child et al. (2019) reduces the complexity by applying combination of strided pattern and local pattern to the vanilla attention matrix. Longformer (Beltagy et al., 2020) designs fixed diagonal sliding windows combined with global window, and the sliding window pattern can also be extended with dilation to enlarge the receptive field. Zaheer et al. (2020) presents a more powerful and expressive sparse attention mechanism, which combines multiple types of attention patterns and gives a thorough study of sparse attention mechanism. Instead of fixed patterns, Kitaev et al. (2019) and Daras et al. (2020) group the attention computation process into buckets by local sensitive hashing, while Roy et al. (2020) uses mini-batch spherical $k$-means. Nevertheless, *Pattern based methods* can only cope with sequences up to a certain length, and the computational complexity still grows rapidly when the input sequence becomes longer.

**Kernel based method**   When faced with longer input sequences, it is more efficient to directly reduce the complexity of the theoretical calculation method. *Kernel based methods* speed up self-attention by reducing the computation complexity of self-attention from quadratic to linear. Vyas et al. (2020) approximate the full attention with a fixed number of cluster attention groups by assuming neighbouring queries in Euclidean space should have similar attention distributions. Peng et al. (2020) chooses to use the production of Gaussian kernel functions to approximate *Softmax*, changing the order of scale dot product calculation, thus reducing the theoretical time to linear complexity and Choromanski et al. (2020) uses Haar measurement based kernel instead. Wang et al. (2020) imports the low-rank prior for attention matrix and approximate *softmax* with SVD decomposition manner. Xiong et al. (2021) utilizes the Nyström method with segment-means to generate a low-rank approximation of the Softmax matrix. Katharopoulos et al. (2020) formalizes the transformer layer as a recurrent neural network. In this paper, we demonstrate that the approximation to *Softmax* is unneccessary for Linearization of self-attention module. We instead propose a new method to replace *Softmax* with a linear operation with a re-weighting mechanism, which reduces both time complexity and space complexity to $O(N)$ while maintaining the accuracy.

## 5 CONCLUSION

We presented COSFORMER , a new efficient transformer that has linear time and space complexity. Our COSFORMER is based on two key properties of the original softmax attention: (i) every element in the attention matrix are non-negative, such that negatively-correlated information are not included for contextual information aggregation; (ii) the non-linear re-weighting scheme concentrates the distribution of the attention matrix, in order to better exploit the locality inductive biases on sequence modeling. To fulfill these properties in our COSFORMER , we utilized the RuLU function as our linear operation to ensure the non-negative property; a new *cos*-based re-weighting mechanism was proposed to enforce the locality bias in the original softmax attention. Since our COSFORMER is naturally decomposable, it does not suffer the accumulated approximation error that usually happens in previous linear transformers. On causal pre-training, bidirectional pre-training, and multiple downstream text understanding tasks, COSFORMER achieves comparable or even better performances than the vanilla transformer. On long sequence benchmark, COSFORMER achieved state-of-the-art performance over five different tasks. Further, COSFORMER obtains a significant overall advantage in terms of time and memory efficiency over all existing efficient transformers, facilitating the transformers to easily scale to longer input sequence.

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

## A  APPENDIX

### A.1  MATHEMATICAL DERIVATION OF *cos*-BASED RE-WEIGHTING

Following Equation 11, we give a detailed deviation of how to obtain output at position $i^{th}$ position:

$$
\begin{aligned}
O_i &= \frac{\sum_{j=1}^{N} f(Q_i', K_j') V_j}{\sum_{j=1}^{N} f(Q_i', K_j')} \\
&= \frac{\sum_{j=1}^{N} \left( \bar{Q}_i^{cos} \left( \bar{K}_j^{cos} \right)^T + \tilde{Q}_i^{sin} \left( \tilde{K}_j^{sin} \right)^T \right) V_j}{\sum_{j=1}^{N} \left( \bar{Q}_i^{cos} \left( \bar{K}_j^{cos} \right)^T + \tilde{Q}_i^{sin} \left( \tilde{K}_j^{sin} \right)^T \right)} \\
&= \frac{\sum_{j=1}^{N} \bar{Q}_i^{cos} \left( \bar{K}_j^{cos} \right)^T V_j + \sum_{j=1}^{N} \tilde{Q}_i^{sin} \left( \tilde{K}_j^{sin} \right)^T V_j}{\sum_{j=1}^{N} \bar{Q}_i^{cos} \left( \bar{K}_j^{cos} \right)^T + \sum_{j=1}^{N} \tilde{Q}_i^{sin} \left( \tilde{K}_j^{sin} \right)^T} \\
&= \frac{\sum_{j=1}^{N} \bar{Q}_i^{cos} \left( \left( \bar{K}_j^{cos} \right)^T V_j \right) + \sum_{j=1}^{N} \tilde{Q}_i^{sin} \left( \left( \tilde{K}_j^{sin} \right)^T V_j \right)}{\sum_{j=1}^{N} \bar{Q}_i^{cos} \left( \bar{K}_j^{cos} \right)^T + \sum_{j=1}^{N} \tilde{Q}_i^{sin} \left( \tilde{K}_j^{sin} \right)^T}
\end{aligned}
\tag{13}
$$

where $i, j = 1, ..., N, M \geq N$, and $Q' = \text{ReLU}(Q), K' = \text{ReLU}(K)$. Let $Q_i^{cos} = Q_i' \cos\left(\frac{\pi i}{2M}\right)$, $Q_i^{cos} = Q_i' \cos\left(\frac{\pi i}{2M}\right)$, $K_j^{cos} = K_j' \cos\left(\frac{\pi j}{2M}\right)$, $K_j^{sin} = K_j' \sin\left(\frac{\pi j}{2M}\right)$. It presents that the output of the proposed COSFORMER attention can be obtained in a linear manner.

### A.2  PSEUDO CODE OF COSFORMER

Algorithm 1 describe the way to compute COSFORMER attention

---
**Algorithm 1** COSFORMER attention
---
**Input:** $Q \in \mathbb{R}^{N \times d_1}, K \in \mathbb{R}^{M \times d_1}, V \in \mathbb{R}^{M \times d_2}$;
**Output:** $O \in \mathbb{R}^{N \times d_2}$;
Use $M_i$ to represent the $i$-th row of matrix $M$;
**Initialize** $A[i] = \frac{\pi i}{2N}, O[i][j] = 0, i = 1, \ldots, N, j = 1, \ldots, d_2$;
**Initialize** $S^{cos}[i][j] = 0, S^{sin}[i][j] = 0, T^{cos}[i] = 0, T^{sin}[i] = 0, i = 1, \ldots, d_1, j = 1, \ldots, d_2$;
**for** $i$ in $1, \ldots, M$ **do**:
    $K_i^{cos} = K_i \cos\left(\frac{\pi i}{2M}\right), K_i^{sin} = K_i \sin\left(\frac{\pi i}{2M}\right)$;
    $S^{cos} \mathrel{+}= \left(K_i^{cos}\right)^T V_i$;
    $S^{sin} \mathrel{+}= \left(K_i^{sin}\right)^T V_i$;
    $T^{cos} \mathrel{+}= K_i^{cos}$;
    $T^{sin} \mathrel{+}= K_i^{sin}$;
**end for**
**for** $i$ in $1, \ldots, N$ **do**:
    $Q_i^{cos} = Q_i \cos\left(\frac{\pi i}{2M}\right), Q_i^{sin} = Q_i \sin\left(\frac{\pi i}{2M}\right)$;
    $O_i = \frac{Q_i^{cos} S^{cos} + Q_i^{sin} S^{sin}}{Q_i^{cos} T^{cos} + Q_i^{sin} T^{sin}}$;
**end for**
---

### A.3  ALGORITHM TO VISUALIZE ATTENTION MATRIX

Algorithm 2 describe the way to visualize attention matrix as Figure 3

---

**Algorithm 2** Algorithm to visualize attention matrix

---

**Input:** $M_k \in \mathbb{R}^{d \times d}, k = 1, \ldots, n; threshold \in [0, 1];$
**Output:** $M \in \mathbb{R}^{d \times d};$
**Initialize** $M[i][j] = 0, i \in 1, \ldots, d, j \in 1, \ldots, d;$
**for** $k$ in $1, \ldots, n$ **do**:
    **for** $i$ in $1, \ldots, d$ **do**:
        index = argsort($M_k[i]$) (in descending order)
        $p = 0$
        **for** $j$ in $1, \ldots, d$ **do**:
            $l = $ index$[j]$
            $p \mathrel{+}= M_k[i][l]$
            $M[i][l] \mathrel{+}= 1$
            **if** $p > threshold$ **then**: break
            **end if**
        **end for**
    **end for**
**end for**
$M \mathrel{/}= n;$
Use heatmap to visualize M;

---

## A.4 INTRODUCTION OF DATASET

We train both models on autoregressive language modeling and bidirectional modeling by Wikitext-103 dataset, it is split by tokens and its statistics as Table 7.Then we fine-tune the pre-trained bidirectional modeling on several text classification tasks.

QQP dataset contain thousands of sentence pair from community question-answering website Quora.Network need to determine pairs of question are semantically equivalent. SST-2 and IMDB are collections of movie reviews. The task is to determine whether a review is positive or not. AMAZON dataset contains millions of product reviews from Amazon.The requirement of this task is to infer the scoring of the product from the review text.MNLI is a crow-source collections of sentence pairs. The network must distinguish which of the three categories entailment, contradiction and neutral the given sentences belong to.

The long-range-aren benchmark contains 5 different datasets.ListOps contains some designed clever mathematical problem to clarify the parsing ability of neural models. IMDB is also used in this benchmark to examine the text classification ability of neural models. CIFAR-10 is a image collection of various of object, this task require models capture 2D spatial relations between flatten pixels.In pathfinder task, models need to determine the connection of two points in the picture, so as to examine the model's ability to acquire 2D spatial relationships.AAN dataset is used to evaluate the ability for models to encode and store compressed representations for retrieving.

| Data | Train | Valid | Test |
|------|-------|-------|------|
| WikiText-103 | 103M | 218K | 246K |
| QQP | 364K | - | 391K |
| SST-2 | 67K | - | 1.8K |
| MNLI | 393K | - | 20K |
| IMDB | 25K | - | 25K |
| AMAZON | 3M | 168K | 168K |
| ListOps | 90K | - | 10K |
| AAN | 147K | 18K | 17K |
| CIFAR-10 | 50K | - | 10K |
| Pathfinder | 160K | - | 20K |

Table 7: Statistics for the datasets.A subset of "Small" amazon subset on electronics category is used for experiment

A.5   QUALITATIVE RESULTS OF LRA

We provide our qualitative results of the ListOps and Document Retrieval tasks on Long-Range-Arena benchmark (Tay et al., 2020b) with a comparison to the vanilla transformer.

ListOps is a ten-way classification task which aims to prediction the results of a sequence with a hierarchical structure and operators MAX, MEAN, MEDIAN and SUM MOD that are enclosed by delimiters (brackets). The network needs to access all tokens and model the logical structure of the inputs in order to make a prediction.

Document Retrieval task is to decide whether the two input long documents are similar or not with a binary label. This task evaluates a model's ability to encode and store compressed representations that are useful for matching and retrieval. Since the samples in LRA are too long, We substantially shorten some selected samples and display them as below:

Listops:
```
1 Input: ( ( ( ( ( ( ( [MED 7 ) 9 ) 3 ) 1  ...... 5 ) 6 ) 8 ) ] ) ) 2 )
      8 ) 9 ) 5 ) 0 ) ] ) ) 8 ) 5 ) 1 ) 2 ) ] )  Our Output: 0,
      Transformer output: 9, Ground-truth: 0
2
3 Input: ( ( ( ( ( ( ( ( ( [SM 5 ) 6 ) 0 ) 7 ) 1 ) ( ( ( ( ( (...... ( (
        ( Input: ( ( [MIN 5 ) 8 ) 1 ) 0 ) (( [MED ( ( ( 8 ) 7 ) 2 ) 8 ) 1
      ) 8 ) ] ) ) 7 ) ] )] ) Our output: 9, Transformer output: 3,
      Ground-truth: 9
4
5 Input: ( ( ( ( ( ( ( ( ( [MAX 7 ) 4 ) 8 ) ( ( ( ( ( ( ( ( ( ( [MAX 5
      ) 2 ) ( ( ( ( ( [SM 3 ) 6 ) 9 ) ( ( ( ...... ) ) 1 ) 6 ) 4 ) 2
      ) ] ) ) ] ) Our output: 9, Transformer output: 5, Ground-truth: 9
```
Listing 1: Examples of LisOps

Byte-level document retrieval:
```
1 Text1: b'1 Introduction Recent advances in Statistical Machine
      Translation (SMT) are widely centred around two concepts: (a)
      hierarchical translation processes, frequently employing
      Synchronous Context Free Grammars (SCFGs) and (b) transduction or
      synchronous rewrite processes over a linguistic ......
2
3 Text2: b'1 Introduction  Automatic Grammatical Error Correction (GEC)
       for non-native English language learners has attracted more and
      more attention with the development of natural language processing
      , machine  learning and big-data techniques. ?The CoNLL2013 shared
       task focuses on the problem of GEC  in five different error types
       including determiner,  preposition, noun number......
4
5 Our output: False, Transformer output: True, Ground-truth: False
```
Listing 2: Examples of Document Retrieval

