# OpenReview forum: "cosFormer: Rethinking Softmax In Attention"
_ICLR.cc/2022/Conference — ICLR 2022 Poster_

### Official Review · Reviewer_M72S · 2021-11-02

**Correctness:** 3
**Technical Novelty And Significance:** 3
**Empirical Novelty And Significance:** 3
**Recommendation:** 6
**Confidence:** 4

**Details Of Ethics Concerns:**

No ethics concern.

**Main Review:**

**Strengths**

- The paper is well-written. With a clear logic flow, the paper is also easy to understand.

- Two factors for improving the efficiency of transformers: (1) the attention matrix should have non-negative elements and (2) aggregating negatively-correlated information needs the non-linear re-weighting strategy, are reasonable, especially the latter one.

- The proposed cosForm, leveraging ReLU and  the Ptolemy’s theorem (cosine re-weighting), is simple yet new, to the best of my knowledge. Glad to see it shows a good accuracy-efficiency tradeoff when applying it to replace the softmax operator in calculating self-attention for transformers.

**Weaknesses**

- In the Method section, when analyzing the computational complexity for the vanilla self-attention and the linearized design, the authors assume " The input length is N and feature dimension is d, with d << N." However, in the experiments, such an assumption does not well hold. For instance, in section 4.1, the sequence length is 512 while the projected feature dimension is 1024. In this perspective, I am concerned by the claimed motivation/advantage of the proposed method, can "linear space and time complexity" still be well supported?

- The limitations of cosForm should be discussed, particularly a more in-depth discussion of its connections to many existing works is necessary.

- As for the comparison of runtime speed in training and inference on the long-range-arena benchmark, did the authors adopt the same transformer architecture for all methods? Why different methods sometime show distinct trends for runtime speed in training vs. inference, e.g. Linear Trans. vs. cosFormer?

- How about the performance of applying cosFormer to other tasks such as in computer vision besides NLP tasks?

- Some grammar errors or typos:

   on page 2, "Kitaev et al. (2020) group"->"Kitaev et al. (2020) groups"

   on page 4, "a computation complexity of O(Nd)"->"a computation complexity of O(Nd^2)"

   on page 7, "despite Longformer (Beltagy et al., 2020) achieve"->"despite Longformer (Beltagy et al., 2020) achieves"

**Summary Of The Paper:**

This paper presents cosFormer which presents a linear operator to replace the softmax operator in calculating self-attention for transformer architectures, maintaining competitive predication accuracy while enjoying linear space and time complexity instead of quadratic costs compared to the vanilla baseline. Under this context, the authors emperically show that two factors are critical to improve the efficiency of transformers: (1) the self-attention matrix should have non-negative elements and (2) aggregating negatively-correlated information needs the non-linear re-weighting strategy. Based on the insightful analysis, the proposed cosFormer adopts ReLU and the Ptolemy’s theorem (cosine re-weighting) in designing a linear replacement of the softmax operator. The performance of cosFormer is validated with the language modeling (autogressive in both causal and non-causal cases) on WikiText-103 dataset,  the finetuning on a lot of downstream datasets such as GLUE, IMDB and AMAZON, and the comparison with state of the art methods on the long-range-arena benchmark.

**Summary Of The Review:**

Please see the comments in "Main Review".

---

> ### Author Response · Authors · 2021-11-20
> **Response to Reviewer M72S**
>
> We would like to thank you for your valuable comments and fix the pointed grammar errors and typos in the revised version. Below we will address your comments.
>
> 1. Linear space and time complexity
>
> As analyzed in the "Response to Reviewer yo75", the dimension of $d$ is the feature dimension of **one head** rather than the total feature dimension. Since the choice of heads is 16 in most cases, the assumption of $d<<N$ still holds, e.g., 64<<512 in your example. In fact, in NLP tasks, we often use a sequence length of 1024 (GPT 2) or 2048 (GPT3), which can largely leverage the advantage of linear transformers.
>
> 2. Limitation of COSFormer
>
> Thanks for the suggestion, we will add a discussion section for that in the revised version. One limitation of our COSFormer is that since we introduce locality into the efficient transformer, our method may suffer a performance lag than other SOTA methods when the data do not have such locality bias. An example can be the "Pathfinder" task, where the two points can be far away from each other. Our method is around 10% worse than the SOTA method, despite that the performance gap between our method and the vanilla transformer is small (70.33 vs 71.4).
>
> 3. Setting of the Long-range-arena benchmark
>
> Yes, we do adapt the same setting for all methods, including the network architectures, hyperparameters and etc. The speed trends may become different in training and inference as the training involves backpropagation. The computational complexities may be different in backpropagation between Linear Trans. and COSFormer and thus lead to different speed trends.
>
> 4. Performance of our method on CV and other tasks.
>
> We are applying our method on several ViT based frameworks such as PVT-v2 for some dense prediction tasks. Results will be shown in our future work.

---

> > ### Comment · Reviewer_M72S · 2021-11-23
> > **Comments on rebuttal**
> >
> > I would like to thank the authors for providing their responses. I am statified with the responses to my first two concerns, but the other two concerns are not well addressed. I want to hear more in-depth feedbacks but not some conjectures or no sure generalization abilities, which are critical to see what happens in the experiments with the proposed method, helping readers have a better understanding of the proposed method.

---

> > > ### Author Response · Authors · 2021-11-23
> > > **Response to Reviewer M72S**
> > >
> > > Thanks for your kind response. I will address the last two concerns as below.
> > >
> > > 1. Different speed trends between training and inference of our method and the linear transformer.
> > >
> > > As shown in Table 5, our method is slightly faster than the linear transformer in training but slower in inference. Let us start with the inference computation complexities. The linear transformer uses an activation layer of $elu(x)+1$, and the COSFormer uses an activation layer of ReLU and a cosine non-linear reweighting scheme, which will cause more computations in inference. However, in the training phase, the linear transformer stores the matrix of $\phi(K)V^T$  as an internal state and updates it at every time step which will cause extra time in training.  On the other hand,  the COSFormer does not have these steps and therefore it processes faster than the linear transformer in training.
> > >
> > > 2. Performance of our method on CV and other tasks.
> > >
> > > In our preliminary experiments of applying COSFormer on the camouflage object detection with the PVT-v2 backbone, our method can achieve comparable results with the vanilla transformer while enjoying faster processing time and smaller memory consumption.

---

> > > > ### Comment · Reviewer_M72S · 2021-11-29
> > > > **Comments on more responses**
> > > >
> > > > I appreciate the new responses from the authors. These explanations are mostly more reasonable than before, and thus I keep my original recommendation.

---

### Official Review · Reviewer_iBUe · 2021-11-02

**Correctness:** 3
**Technical Novelty And Significance:** 3
**Empirical Novelty And Significance:** 3
**Recommendation:** 8
**Confidence:** 5

**Main Review:**

Strengths:
- The proposed method is simple and intuitive and experimentation on both LRA and real world language modeling task shows that it provides better speed-accuracy trade-offs when compared against other effective attention mechanisms.
- The paper is well written and easy to follow.

Some questions and concerns:

- Related work section could be expanded to include discussion on other efficient attentions such as:
    - Approximating softmax atttention:
        - SMYRF: Efficient Attention using Asymmetric Clustering
        - Nyströmformer: A Nyström-Based Algorithm for Approximating Self-Attention
        - Fast Transformers with Clustered Attention
    - Sparsity pattern
        - Efficient Content-Based Sparse Attention with Routing Transformers
    - Low rank projections:
        - Set Transformer: A Framework for Attention-based Permutation-Invariant Neural Networks
- Related work on the properties for kernelized attention (non-negative similarity scores):
    - Transformer Dissection: A Unified Understanding of Transformer's Attention via the Lens of Kernel

- Clarification:
    - For bidirectional modeling, why are the models pre-trained only to 50K iterations? From figure 4, the validation perplexity still seems to be decreasing especially for the vanialla attentions.
    - In the table 1, is $\phi_{ReLU}$ linear attention with ReLU as the non-linear activation to get positive similary scores?
    - Similarly for table 6, without re-weighing is linear attention with ReLU activations?

- Relation with relative positional encodings for linear attentions:
    - The cosine re-weighing can also be looked as an efficient way to introduce relatve positional bias to the linear attention. In this regard it would be good to contrast this to recently introduced relative positional encoding for linear attention such as (a) Rotary Position Embedding and (b) Relative Positional Encoding for Transformers with Linear Complexity

- Page 5: "The non-linear re-weighting mechanism introduced by the softmax attention can concentrate the distribution of the attention weights and therefore stabilize the training process. It also means that, to some extent, it will punish far-away connections and enforce locality." While this is empirically observed, softmax normalization doesn't introduce such inductive biases. It might be better to rephrase this differently.


**Summary Of The Paper:**

This work introduces a simple and elegant way to incorporate recency bias for kernelized linear attention mechanism using cos-based re-weighting mechanism.  This cosine re-weighting can be effectively implemented via query and key matrix transformations for linear attention mechanisms. The authors experiment with LRA, and both language modeling with autoregressive and bidirectional setups to show the efficacy of proposed method.

**Summary Of The Review:**

This work introduces a simple mechanism to add relative attention / locality bias to linear attention to significantly improve their performance. The results showcase that using the proposed cosine re-weighting the linear attention achieves similar performance to vanilla transformers while being significantly faster.

---

> ### Author Response · Authors · 2021-11-20
> **Response to Reviewer iBUe**
>
> Thank you for your valuable comments especially for the latest related work, we will add them into the related work section in the revised version. We will address your concerns below.
>
> 1.  Pre-training iterations.
>
> In our preliminary experiments, we find that the validation perplexity does not change much when training longer than 50K iterations. Also, the model with a longer training time does not show better fine-tuning results on the downstream tasks such as QQP, SST-2, and MNLI. Therefore, we use 50K as the total pre-training iterations for all model variants.
>
> 2. $\Phi_{ReLU}$.
>
> As shown in the "Response to Reviewer ZpU1", the $\Phi_{ReLU}$ is just added a ReLU activation to $Q, K^T$ features to make sure the similarity scores are non-negative.
>
> 3. "w/o reweighting" in Table 6.
>
> Sorry for the confusion, the variant of "w/o reweighting" is identical with $\Phi_{ReLU}$. We will unify these annotations to avoid confusion in the revised version.
>
> 4. Relation with relative positional encodings for linear attentions.
>
> We agree that our COSFormer can be seen as a new way of introducing the relative positional bias to the efficient transformer. Compared with the Rotary Position Embedding, they use a much more complex position embedding strategy and did not enforce the non-negativity to the similarity scores. Also, since they only change the position embedding on the numerator while keeping the denominator unchanged,  the summation of their attention scores is not equal to 1. For the Relative Positional Encoding for Transformers with Linear Complexity, they use sampling strategy to approximate the softmax, which is more similar to RFA. We will add a detailed discussion in the revised version.
>
> 5. Rephrase the sentence.
>
> Sure, we will rephrase the sentence as "we empirically find that the non-linear re-weighting mechanism introduced by the softmax attention concentrates the distribution of the attention weights and to some extent, punish far-away connections and enforce locality".

---

### Official Review · Reviewer_ZpU1 · 2021-11-03

**Correctness:** 2
**Technical Novelty And Significance:** 3
**Empirical Novelty And Significance:** 3
**Recommendation:** 6
**Confidence:** 5

**Main Review:**

## Strengths

1. The proposed method accelerates the inference speed of the model.

## Weakness

1. The motivation of using ReLU is unclear. Though softmax promises a non-negative weight, there is no evidence showing that non-negative weights is essential.

2. The motivation of cosine operation is unclear. The success of self-attention partly attributes to its ability of building long-distance dependency. The authors introduce locality into self-attention through consine operation but do not show its rationality.

3. Lack of essential ablation studies. How about the model without ReLU?

4. Limited improvement. In terms of Table 6, the performance gain induced by the cosine operation is limited (~0.13) on some models.

**Summary Of The Paper:**

The paper proposes a substitute of the vanilla self-attention module, which claims a linear complexity. Like Performer and many other previous works, it introduces kernel functions and change the computation order among QKV in the self-attention module. Essentially, it replaces softx with ReLU and a cosine operation. Experiments are done on several benchmarks.

**Summary Of The Review:**

Totally, I think the paper is still not well-prepared for publication.

---

> ### Author Response · Authors · 2021-11-19
> **Response to Reviewer ZpU1**
>
> We thank the reviewer's valuable comments. Below, we address the main concerns by quoting the comment followed by our response.
>
> 1. Motivation for using ReLU and why non-negative weights are essential.
>
> We justified our selection of ReLU in Table 1, where the $\phi_{\mathbf{I}}$ represents the variant without non-negativity, and $\phi_{\mathrm{ReLU}}$ is the ReLU variant. By introducing the non-negativity into the attention, the performance/accuracy increased 14.9% in the validation loss, 5.0% in QQP, 17.9% in SST-2, and 33.6% in MNLI, which empirically demonstrated the essence of non-negativity.
>
> 2. Motivation of cosine operation.
>
> Introducing the non-linear re-weighting scheme into the efficient transformer is the key insight of this work, where previous methods often neglect this property of the vanilla SoftMax. The self-attention mechanism does have the capability to capture the global dependency, but it does not mean that it will assign a similar attention weight to every position. In fact, as shown in Figure 3, most of the attention weights are assigned to neighboring positions (the diagonal of the matrix) in real-world experiments. Inspired by this observation, we propose the cosine operation to mimic this property. Note that not all non-linear re-weighting operations can be used in the efficient transformer as the operation has to be decomposable such that the Eq. (5) can be held. Also, our extensive experiments on autoregressive language modeling, bidirectional language modeling, and long-range-arena benchmark validate the effectiveness of our method.
>
> 3. Model without ReLU.
>
> We provide this model's result in Table 1 (refer to $\phi_{\mathbf{I}}$ as the model without ReLU).
>
> 4. Limited improvement.
>
> As shown in Table 2, Table 3, our method beats all other competitors with a clear margin in autoregressive language modeling, bidirectional language modeling. We also show superior speed and accuracy balance on the long-range-arena benchmark in Figure 1, even better than the vanilla transformer in terms of accuracy.  Considering the cosine operation alone, the importance of locality may vary from task to task, but the more important thing is that the cosine operation can constantly improve the performance of efficient transformer in most of the tasks.

---

> > ### Comment · Reviewer_ZpU1 · 2021-11-19
> > **The choice of cosine is an interesting operation**
> >
> > From the perspective of imitating softmax, the authors have convinced me that the cosine operation is an interesting operation, and results also show its effectiveness.
> >
> > You may wish to see the new comments for other concerns.

---

> ### Comment · Reviewer_ZpU1 · 2021-11-19
> **Re-claim about my reviews**
>
> I ever doubted my opinions after knowing other reviewers' ratings, so I read the paper carefully again, while I still think this paper has not been well-prepared for publication. I'll supplement my review as follows:
>
> The proposed cosFormer achieves good performance on several datasets such as GELU, IMDB, etc. The results show that the proposed method is effective in practice. Further, the authors's response have convinced me that the choice of cosine is interesting, and it does smooth the attention matrix.
>
> However, I cannot agree with some claims in the paper and think the model may work in a different way from the descriptions in the paper. The specific concerns are as follows:
>
> 1. The comparison between ReLU and Identity (in Table 1) cannot prove the importance of non-negativeness because ReLU also brings non-linearty to the model. I'll give two examples for further explaination:
>    1. Take a model with ReLU as the only activation, and replace all ReLU as Identity function. In this case, the model will perform bad because it becomes a linear model. Does it prove that non-negativeness is important to this model?
>    2. For comparison, the authors should at least test several other activation functions. For example, LeakyReLu may output a negative value. Will LeakyReLU lead to a result similar to Identity? Or, more experiments may tell us that Transformer does not even need non-negativeness, and many activation functions work well.
>
> 2. The choice of ReLU is heuristic. There are many activations that promise a non-negative output (e.g., Sigmoid). Why not others but ReLU?
>
> I will temporarily keep my rating.
>
> Totally, the experiments in this paper are interesting and can enlighten us for further exploration. However, I do not think the existing contents can defend the claims in the paper. It sill needs more analysis and test to yield the conclusion.

---

> > ### Author Response · Authors · 2021-11-19
> > **Response to Reviewer ZpU1**
> >
> > Sorry for the confusion in Table 1.  I will explain in detail about this table.
> >
> > Given a general form of attention $$\mathcal{O}_i= \sum_j \frac{\mathcal{S}(Q_i,K_j)}{\sum_j \mathcal{S}(Q_i,K_j)}V_j,$$
> >
> > in the vanilla SoftMax, we have $${\mathcal{S}_E(Q,K) = \exp(QK^T)},$$
> >
> > In our $\phi_{\mathbf{I}}$, we have $${\mathcal{S}_I(Q,K) = QK^T},$$
> >
> > and for $\phi_{\mathrm{ReLU}}$, we have $${\mathcal{S}_{R}(Q,K) = ReLU(Q)ReLU(K^T)}$$
> >
> > In this case, we do not "take a model with ReLU as the only activation, and replace all ReLU as Identity function", rather add the $\phi_{\mathbf{I}}$ with a ReLU activation function to make sure all attention weights are non-negative.
> >
> > However, we agree with the reviewer that we can add the LeakyReLU to Table 1 for the sake of completeness. We show the result of LeakyReLU beblow. We will update this result in the revised version.
> >
> > | Variant             | Loss  | QQP   | SST-2 | MNLI  |
> > |---------------------|-------|-------|-------|-------|
> > | $\phi_{\mathbf{I}}$ | 2.343 | 84.23 | 76.26 | 58.27 |
> > | LeakyReLU           | 2.246 | 84.46 | 78.21 | 74.26 |
> > | ReLU                | 1.993 | 88.86 | 89.90 | 77.86 |
> >
> > From this result, we can find that the performance of the LeakyReLU is very close to $\phi_{\mathbf{I}}$ in the Loss, QQP, and SST-2 and has a large gap in MNLI compared with the ReLU variant. It further proves our assumption of non-negativity in attention.
> >
> > For the choice of other activations, we did validate other variances such as Sigmoid but it did not show comparable results as ReLU.

---

> > > ### Comment · Reviewer_iBUe · 2021-11-19
> > > **Thoughts on Non-negativity, ReLU, and other choices**
> > >
> > > The contribution regarding the importance for non-negative kernel is not necessarily new and has previously also been shown in [1] though with limited experimentation as well. Please take a look at Table 2.
> > >
> > > Based on [1] and current work, I think that non-negative kernel is important. However, I also am similarly concerned about the choice of ReLU being heuristic. I am not convinced by the explanation that Sigmoid did not work because it is a bounded function. In [2], the authors use ${\text{elu(x)} + 1}$ as the unbounded non-negative activation which gets significantly worse scores on LRA: *50.55* vs *54.2* (Table 6). Thus I would agree that adding results for several choices would add to the understanding as well as defend the heuristic choice.
> > >
> > > [1] Transformer Dissection: A Unified Understanding of Transformer's Attention via the Lens of Kernel.
> > >
> > > [2] Transformers are RNNs: Fast Autoregressive Transformers with Linear Attention

---

> > > > ### Author Response · Authors · 2021-11-19
> > > > **Response to Reviewer iBUe**
> > > >
> > > > Thanks for your valuable comments and we share the same opinion that non-negativity matters.
> > > >
> > > > Here, we would like to emphasize that the main contribution of this work is to introduce a new non-linear re-weighting scheme into the efficient transformer rather than the selection of non-negative kernels as it has been partially explored in previous literature.  We are also interested in why the sigmoid function does not work well in this case. The bounded function explanation in the previous comment is only one of our assumptions and has not been validated yet. Compared with [2], the sigmoid one has a validation loss of 7.255 while the [2] is 2.094. Since it is not in the main scope of this paper, we would like to postpone the discussion to our future work.  However, for the choice of ReLU, we will add a comparison of different choices for non-negative kernels in the revised paper, i.e., LeakyReLU, Sigmoid, ${elu(x)}+1$.

---

> > > ### Comment · Reviewer_ZpU1 · 2021-11-23
> > > **The new responses resolve more concerns from me**
> > >
> > > Thanks for the authors' careful responses.
> > >
> > > I think the authors' new responses resolve more concerns from me. I now acknowledge that non-negativity is an important property of self-attention (also thanks the remind from Reviewer iBUe).
> > >
> > > Though I still hope more explainations about the the choice of ReLU, I am willing to raise my rating.

---

### Official Review · Reviewer_yo75 · 2021-11-05

**Correctness:** 4
**Technical Novelty And Significance:** 3
**Empirical Novelty And Significance:** 3
**Recommendation:** 8
**Confidence:** 3

**Main Review:**

Strengths
+ The paper is very well written and organized. It explains the problem and the proposed solution clearly.
+ The authors have been thorough in experimental analysis, and have shown the results on benchmarks as well as the generalization ability of the method.
+ The idea of reweighing with a cos based function is faily novel.

Weaknesses
- In Section 3.2,  it is mentioned that "in typical natural language tasks, the feature dimension of one head is always much smaller than the
 input sequence length N". What implications it could have when d is not that small. Would the analysis will still be applicable (especially the computational speedup for domains such as images) ?
-  In Table 4, results on "Pathfinder", the method significantly lags behind the state of the art (~10%). An explaination would help understand the challenges of this particular setting.
- As the method is shown to significantly outperform state of the art on a few benchmarks, a few examples on where it performs better while the other methods perform inferior should be provided to qualitatively analyse the improvement.
- A few typos (i) Section 3.1 "...the Eq. 2 become" -> Eq. 2 becomes (ii) Section 3.4, "...such locality bias, ie.,..." -> i.e. (note the dots)


**Summary Of The Paper:**

The paper proposes a variant of the transformer network. The authors base their work on their analysis of softmax attention. They argue that softmax works well because of two reasons (i) it stabilizes training due to reweighing the attention as well as associated connections within the network, and (ii) it forces non negative values in the attention matrix. Utilizing these observations, they propose two modifications (i) a linear projection kernel i.e. ReLU to compute similarity, and (ii) a cosine function for reweighing the attention values. With exhaustive experiments on 5 benchmarks, authors demonstrate that the proposed modifications lead to state of the art performance with reduced computational compelxity.

**Summary Of The Review:**

The paper is generally well written and the problem is motivated clearly. The paper also highlights contribution with reference to the compared methods. The experiments (empirical) are exhaustive, however, a few qualitative results on explanations of a few disparities in the performance of the proposed method (as indicated in the detailed review) would help understand the impact of the proposed method better.

---

> ### Author Response · Authors · 2021-11-20
> **Response to Reviewer yo75**
>
> Thanks for your valuable comments. Below, we address the main concerns as follows.
>
> 1. What will happen when $d$ is not that small.
>
> In Figure 2, we show the computation complexities of a vanilla self-attention as $O(N^2d)$ and a linear one as $O(Nd^2)$. As long as $N>d$, the linear attention will show its advantages in speed. It is worth noting that the dimension of $d$ is the feature dimension of **one head**.  In this case, when "the sequence length is 512 while the projected feature dimension is 1024"(as quoted from Reviewer M72S), the size of $d$ should be $1024/16 = 64$ if we use 16 heads and the assumption of $N>d$ holds.
> Also, thanks to the memory efficiency, linear attention allows us to use longer sequences as input, i.e., 4096 or longer, which can not be done with vanilla attention due to the current hardware constraint.
>
> 2. Explanation of the "Pathfinder" result.
>
> The Pathfinder requires the network to decide whether the two points represented as circles are connected by a path consisting of dashes.
> The distance between these two points will vary randomly: they can be very far from each other. However, since we introduce locality into the efficient transformer, our method would pay more attention to the nearby positions and thus show a bit lags to other SOTA methods.
>
>
> 3. Qualitative results
>
> Thanks for the suggestion, we will add some qualitative results of the LRA benchmark in the revised version, such as listops and retrieval tasks.
>
> 4. Typos
>
> Thanks for pointing them out, we will revise them carefully in the revised version.

---

### Public Comment · ~Shuhao_Cao1 · 2022-01-29
**Missing references**

The cosine reweighting scheme is like a change of basis (like DFT).

[1]: Guibas et al, Efficient Token Mixing for Transformers via Adaptive Fourier Neural Operators.

[2]: Lee-Thorp et al, FNet: Mixing Tokens with Fourier Transforms.

---

> ### Public Comment · ~Yiran_Zhong1 · 2022-02-06
> **RE: Missing reference**
>
> Many thanks, Shuhao. We will add these two references in our camera-ready version with a discussion.
>
> However, the intuitions behind between ours and these two papers are different. Ours is based on the locality assumption and happens to use a cosine re-weighting scheme while these two are based on DFT.
>
> Note that in FNet, there are no learnable parameters in Fourier transforms and we have trainable parameters for the Q, K, Vs. For [1], it is an ICLR 2022 paper and it improves the FNet by making the Fourier transform trainable.

---

### Decision · Program_Chairs · 2022-01-20

**Decision:**

Accept (Poster)

**Comment:**

This paper introduces a new linear attention mechanism for transformer based models.  This is accomplished by replacing the softmax in the standard transformer self-attention with a cosine-based re-weighting mechanism.  The empirical results are good, and cosFormer generally outperforms existing efficient transformers for autoregressive language modeling, fine-tuning, and on the long range arena.

The reviewers were generally positive regarding the paper, with all reviewers voting to accept.  The discussion period focused on particular choices regarding the ReLU activation function vs. other non-negative activation functions, further motivating the cosine operation, and comparing the speed of cosFormer vs. other efficient transformers.  The authors responded by providing additional ablations to empirically validate the choice of ReLU, motivated the cosine operation by noting that it introduces a locality bias, and further described the computation requirements of their transformer vs. prior work.

Overall, this is an interesting addition to the linear / efficient transformer literature, with solid empirical results supporting the various design decisions.